# STORYANALOGY: Deriving Story-level Analogies from Large Language Models to Unlock Analogical Understanding

**Cheng Jiayang**♠◇ **Lin Qiu**◇ **Tsz Ho Chan**♠ **Tianqing Fang**♠ **Weiqi Wang**♠
**Chunkit Chan**♠ **Dongyu Ru**◇ **Qipeng Guo**◇ **Hongming Zhang**♠
**Yangqiu Song**♠ **Yue Zhang**† **Zheng Zhang**◇

♠The Hong Kong University of Science and Technology
†Westlake University ◇Amazon AWS AI

{jchengaj, yqsong}@cse.ust.hk zhangyue@westlake.edu.cn zhaz@amazon.com

## Abstract

Analogy-making between narratives is crucial for human reasoning. In this paper, we evaluate the ability to identify and generate analogies by constructing a first-of-its-kind large-scale story-level analogy corpus, STORYANALOGY, which contains 24K story pairs from diverse domains with human annotations on two similarities from the extended Structure-Mapping Theory. We design a set of tests on STORYANALOGY, presenting the first evaluation of story-level analogy identification and generation. Interestingly, we find that the analogy identification tasks are incredibly difficult not only for sentence embedding models but also for the recent large language models (LLMs) such as ChatGPT and LLaMa. ChatGPT, for example, only achieved around 30% accuracy in multiple-choice questions (compared to over 85% accuracy for humans). Furthermore, we observe that the data in STORYANALOGY can improve the quality of analogy generation in LLMs, where a fine-tuned FlanT5-xxl model achieves comparable performance to zero-shot ChatGPT.[1]

## 1 Introduction

Analogy-making plays a central role in human reasoning abilities. By drawing similarities between seemingly unrelated concepts (e.g., in Figure 1, "virus" v.s. "burglar") and processes ("the virus invades cells" v.s. "the burglar breaks into the house"), we can infer that the virus infiltrates and damages cells in a similar way to how a burglar breaks into a house to steal or cause harm. These story-level analogies, which involve comparing entire narratives or coherent sequences of events, enable intelligent agents to gain insights (Boden, 2009; Ding et al., 2023; Bhavya et al., 2023) and understand complex phenomena (Webb et al., 2022).

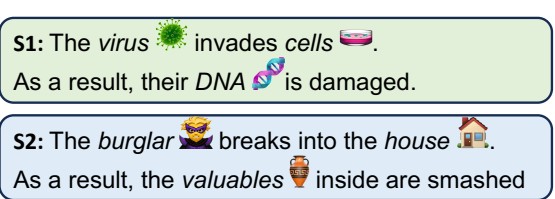

S1: The *virus* invades *cells*.
As a result, their *DNA* is damaged.

S2: The *burglar* breaks into the *house*.
As a result, the *valuables* inside are smashed

Figure 1: An example of analogy between story **S1**: the invasion of cells by a virus, and **S2**: a burglar breaking into a house.

Despite its significance, there has been limited research on story analogies. One of the reasons is the lack of available data and evaluation benchmarks. In contrast, the community has predominantly focused on word-level analogies, which involve identifying relational similarities between pairs of concepts (e.g., *king* to *man* is like *queen* to *woman*) (Mikolov et al., 2013; Gladkova et al., 2016; Czinczoll et al., 2022).

In this work, we introduce STORYANALOGY, a large-scale story-level analogy corpus derived from various domains: scientific scripts, social narratives, word analogies, and knowledge graph triples, to facilitate the study of complex analogies. The story-level analogies we examine contain richer relational details, such as relations between entities (e.g., virus, *invades*, cells) and between events (e.g., the virus invades cells, *as a result*, the virus damages DNAs).

One of the challenges in building STORYANALOGY is establishing a clear and specific way to evaluate story analogies. To address this problem, we extend the Structure-Mapping Theory (SMT; Gentner, 1983) to evaluate on longer texts. According to SMT, analogies hold (e.g., the *hydrogen atom* vs. the *Solar System*) because of the similarity in *relational* information (e.g., the relative motion between objects), rather than *attributive* information (e.g., size), between the source and target. Conversely, if both types of information are similar, the source and target

---

[1]This work was done when Jiayang was an intern at Amazon AWS AI Lab. Code and data are released at: https://github.com/loginaway/StoryAnalogy.

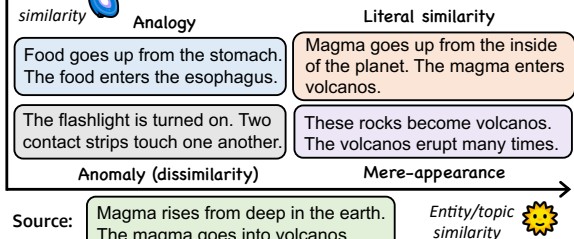

Figure 2: The similarity space, showing different kinds of matches in terms of the degree of relation similarity 🔵 versus entity similarity 🌞 . According to SMT, we can classify the type of matches (Analogy, Literal similarity, Anomaly, and Mere-appearance) between the source and target story by the two similarities. The figure is an extension with story examples based on the visualization in Gentner and Markman (1997).

exhibit a literal similarity (e.g., the *X12 star system* v.s. the *Solar System*). Inspired by this notion, we extend SMT to the story level (§ 2.1). We use *entity* and *relation similarity* to assess the level of similarity in *attributes* and *relations* between the source and target stories. Additionally, we propose an *analogy score* based on these two similarities to quantify the degree of analogy between stories. Figure 2 provides a visual representation of the similarity space spanned by the two similarities.

We then collect candidate story analogies for similarity annotations. Since story analogies are scarce in free texts[2], we use large language models (LLMs) to generate story pairs that are likely to be analogies. The stories are sourced from various domains, including scientific scripts (Dalvi et al., 2018), social commonsense stories (Mostafazadeh et al., 2016), word-level analogies (Turney et al., 2003; Czinczoll et al., 2022), and knowledge graphs (Speer et al., 2017). Next, we conduct crowd-sourcing to obtain similarity annotations for each candidate story pair. As a result, we create STORYANALOGY, which consists of 24K diverse story pairs, each with human annotation guided by the extended SMT.

Based on STORYANALOGY, we curate a set of tests to evaluate the analogy identification ability of models. Our findings indicate that both competitive encoder models (such as SimCSE (Gao et al., 2021) and OpenAI's text-embedding-002) and LLMs (such as ChatGPT (OpenAI, 2022) and LLaMa (Touvron et al., 2023)) have a significant

---

[2]Analogies are only present in approximately 3% of a scientific corpus (Sultan and Shahaf, 2023), and the prevalence is expected to be even lower in general texts.

gap compared to human performance in terms of predicting the level of analogy between stories. We further evaluate LLMs using multiple choice questions derived from the story candidates. Even the best-performing LLM still falls short of human performance by 37.7%. Furthermore, we discover that using stories in STORYANALOGY can enhance models' ability to identify and generate analogies. By employing few-shot in-context learning and finetuning on STORYANALOGY, baseline models achieve a considerable performance boost. For instance, a fine-tuned FlanT5-xxl model exhibits generation quality on par with zero-shot ChatGPT. We hope that the data and evaluation settings we proposed in this study will benefit the research community in the area of story analogies.

## 2 STORYANALOGY

Conventional benchmarks in computational analogy primarily focus on word-level analogies (e.g. *word* to *language* is like *note* to *music*). However, less attention has been given to more sophisticated analogies. We introduce STORYANALOGY, a dataset of 24,388 pairs of stories (e.g., *"The virus invades cells and DNAs are damaged."* versus *"A burglar breaks into the house and smashes the valuables inside."*), each annotated with two dimensions of similarity based to SMT.

### 2.1 Evaluating story analogies

To assess the degree of analogy between a pair of instances, recent studies classify story pairs using a set of labels. For instance, Sultan and Shahaf (2023) use 5 labels including not-analogy, self-analogy, close-analogy, far-analogy, and sub-analogy. Nagarajah et al. (2022) use 6 labels: shallow attribute analogy, deep attribute analogy, relational analogy, event analogy, structural analogy, and moral/purpose. However, they observed very poor agreement among annotators for most labels, which indicates a vague understanding of the task. Making comparisons across these studies are challenging due to the vastly different settings.

In cognitive psychology, the Structure Mapping Theory (SMT; Gentner, 1983) is well-known for its explanation of the cognitive process of making analogies between objects. SMT evaluates object comparisons from two perspectives: (a) the attributes of objects and (b) the relational structures between objects. Analogies between objects

| Source story | Target story | Scores 🧑 | Domain |
|---|---|---|---|
| The stream becomes a river. The river continues to flow along the same path for a long time. | A person grows from a child into an adult. As time passes, the person experiences ongoing growth and maturation. | 🌼: 0.6 🔵: 2.8 | PP |
| They left him the key to the entrance. When Tom went over he realized it was the wrong key. | They gave her the password to the website. When Jane logged in, she realized it was the wrong password. | 🌼: 1.0 🔵: 2.7 | ROC |
| Foundations are poured to support the walls and roofs of buildings. The structure of the building is only as strong as it's foundation. | Reasons are formulated to make theories. The conclusions of theories are only as dependable as their initial premises. | 🌼: 0.6 🔵: 1.8 | WA |
| His memory has broken into fragmented pieces. He can recall flashes and images of the past, but nothing concrete or clear. | His memories remain a confused mess. Nothing holds together and what he remembers don't make sense. | 🌼: 2.7 🔵: 3.0 | WA |
| The student opens the book and begins to read. The knowledge gained from the book is absorbed by the student. | The cat sees a mouse and begins to chase it. The cat honing its hunting skills through practice and repetition. | 🌼: 0.8 🔵: 1.4 | CN |

Table 1: Examples in STORYANALOGY with annotations from each domain. We report the EntSim 🌼 and RelSim 🔵 from crowd workers 🧑 . The **Domain** column indicates the source of the story pairs. "PP", "ROC", "WA", and "CN" are short for "ProPara", "ROCStories", "Word Analogy", and "ConceptNet", respectively.

occur when they have similar relational structures but dissimilar attributes (e.g., the *hydrogen atom* v.s. the *Solar System*). In contrast, literal similarity occurs when objects have both similar relational structures and attributes (e.g., the *X12 star system* v.s. the *Solar System*).

Based on SMT, we propose to compare stories by their *entity* and *relation similarity*. These measures assess the degree of similarity in terms of attributive and relational structures, respectively. We provide necessary extensions to their definitions:

**Entity similarity** (EntSim 🌼 ). The similarity of entity and topics discussed between a pair of stories, ranging from 0 (unrelated) to 3 (almost equivalent). This score should be high if the two stories are both discussing apples and pears, even if they differ greatly in the details.

**Relation similarity** (RelSim 🔵 ). The similarity of relational structures between a pair of stories, ranging from 0 (very poor alignment) to 3 (alignment). In this context, the relational structures refer to the connections between elements at different levels. For instance, first-order relations can be regarded as the relationship between entities, such as predicates. Second-order relations, on the other hand, represent connections between higher granularity elements, such as the logical connection between events or sentences. We encourage annotators to also consider higher-order relational similarity, such as the moral or purpose behind the stories.

We present the established similarity space with example source and target stories in Figure 2.

**Modeling the *analogy score* ($\alpha$).** We discuss possible definitions of the *analogy score* ($\alpha$). The score $\alpha$ should be proportional to the level of analogy between a pair of stories. Defining $\alpha$ to be equivalent with RelSim has been adopted in word analogy (Ushio et al., 2021a). However, this definition cannot distinguish analogy from literal similarity, as both of them have high RelSim (Figure 2). We can alleviate this problem by introducing EntSim to the definition of $\alpha$: according to SMT, analogy happens when the RelSim between the source and target story is high and the EntSim is low[3]. Therefore, in the rest of this paper, we define $\alpha$ as RelSim/EntSim[4].

## 2.2 Distilling story analogies from LLMs

Obtaining a large number of story analogy by retrieval is difficult. Evidence from Sultan and Shahaf (2023) shows that the prevalence of analogies within a categorized dataset is around 3%. It is expected that the ratio is much lower in general corpora. Identifying analogies by retrieving from general corpora would thus require huge human efforts, making it unrealistic to build a large-scale story

---

[3]"An analogy is a comparison in which relational predicates, but few or no object attributes, can be mapped from base to target." (Gentner, 1983)

[4]In practice, we compute it by RelSim/(1+EntSim) to ensure numerical stability.

analogy collection in this way. Recently observations suggest that LLMs are capable of understanding and predicting analogies for problem-solving (Webb et al., 2022), cross-domain creativities (Ding et al., 2023), and generating explanations for word analogies (Bhavya et al., 2022). In addition to these findings, we discover that LLMs can generate high-quality story analogies (i.e., with more than a half generations being analogies). Here, we introduce the pipeline for generating story analogies. The generated analogies are further annotated by crowd annotators for verification. (Details are in § A.1.)

**Curating seed examples.** The first step is to curate a seed set of story analogies. We ask experts from our team to write story analogies. To ensure diversity, the experts are required to consider multiple domains and are allowed to search in corpora or on the Internet. They then determine whether these story pairs are indeed analogies Examples that are not considered analogies are removed from the gold set. As a result, we obtained a total of 28 story analogy examples, each containing a pair of stories and the corresponding entities.

**Source data.** To guarantee the coverage of topics, we sample from corpora of four domains to generate stories, including (1) scientific scripts ProPara (Dalvi et al., 2018), (2) social commonsense stories ROCStories (Mostafazadeh et al., 2016), (3) word analogy evaluation sets[5] SAT (Turney et al., 2003), U2 and U4[6], and SCAN (Czinczoll et al., 2022). , and (4) the commonsense KG ConceptNet[7] (Speer et al., 2017). Note that, source data (1) and (2) consist of stories, while (3) and (4) consist of word pairs.

**Generating story candidates.** Using the seed examples and source data, we prompt LLMs[8] to generate analogies. Due to the different formats of the source data, the story pairs are generated using two different paradigms (Details are in § A.1.):

*Generating from story pairs.* Given a source story and several source-target story pairs sampled from the seed examples, we prompt an LLM to generate

the target story.

*Generating from word pairs.* Given a word analogy pair (e.g. *"word"*, *"language"* and *"note"*, *"music"*), together with source-target analogies with the corresponding entities from seed examples, an LLM is prompted to generate both the source and target stories.

## 2.3 Annotation

To evaluate each candidate story pair under the extended SMT, we conduct crowd annotations on Amazon Mechanical Turk[9]. We recruit crowd workers to annotate the entity and relation similarities for the collected pairs. In addition, workers are required to label an instance as "poor quality" if they find the generated content broken or toxic. The annotation consists of the following two rounds:

**(i) Qualification round.** We first annotate 80 candidate story pairs (20 from each domain) to curate a qualification set. Three domain experts from our team are asked to read through the annotation instruction and independently annotate EntSim and RelSim for these pairs. The Spearman's $\rho$ between each annotator's prediction with the average scores of the others ranges from 93% to 96% on EntSim, and from 89% to 95% on RelSim.

We invite crowd workers who have ≥90% history approval rates and have ≥1K HITs approved to attend the qualification. Workers whose predictions achieve ≥70% Spearman's $\rho$ with the average scores from three experts pass the qualification. As a result, 158 and 80 workers passed the qualification for EntSim and RelSim, respectively.

**(ii) Main round.** Qualified crowd workers are invited to attend the main round annotations. We assign 5 different annotators to give predictions for each similarity of a story pair. To guarantee the annotation quality, we follow the annotation setting in (Agirre et al., 2012). We split the main round into multiple mini-rounds, each with 1K-2K candidate pairs. After each mini-round, we filter out and disqualify workers who do not show significant correlations with the average scores of the others. They are paid more than what is required by the local wage law. In addition, experts from our team manually check the quality of annotations and write feedback to workers correspondingly.

The generated contents sometimes contain hallucinations or toxic contents. We filter out story pairs labeled as "poor quality" by more than 10% an-

---

[5]After manual inspection of all word analogy datasets, we do not include classic datasets such as Google (Mikolov et al., 2013), where the relations between words are relatively easier syntactic or shallow semantic relations, such as ("similar: similarly", "rare: rarely").

[6]https://englishforeveryone.org/Topics/Analogies.html

[7]We consider entity pairs in triples that share the same relations from ConceptNet. https://huggingface.co/datasets/relbert/analogy_questions

[8]OpenAI's text-davinci-003 is used in the generation.

[9]https://www.mturk.com/

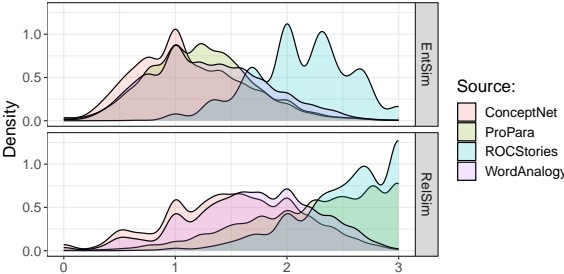

Figure 3: Distributions of `EntSim` and `RelSim` on four data domains in STORYANALOGY. Notably, the distributions of `EntSim` and `RelSim` on ROCStories tend to skew towards higher values. This could be attributed to the fact that stories from this source primarily revolve human-focused social narratives.

notators, which accounts for 142 instances. For each story pair, we adopt the average scores from workers as the predicted `EntSim` and `RelSim`.

## 2.4 Analysis of STORYANALOGY

To assess inter-annotator agreement, we randomly sampled 1K instances with 3 independent annotations from our dataset. The Fleiss's kappa (Fleiss, 1971) on the binarized annotations of `EntSim` are 47%, and 42% on `RelSim`, indicating moderate agreement among annotators. In addition, we additionally obtained expert annotations on 200 randomly sampled instances. The averaged Spearman's correlation between crowd and expert annotations on `EntSim` and `RelSim` is 64.7% and 69.9%, respectively.

The final dataset consists of 24,388 story pairs on four domains: ProPara (6.9K), ROCStories (4.9K), Word-Analogy (7.5K), and ConceptNet (5.0K). Stories in STORYANALOGY have 19.94 tokens on average. The distributions of `EntSim` and `RelSim` are presented in Figure 3. We randomly select 500 instances from each domain as the test set, and another 500 instances from each domain as the validation set. Examples of STORYANALOGY are shown in Table 1.

## 3 Story Analogy Identification

We begin by assessing the ability of models to *identify* story analogies using two different setups. The first evaluation setup is similar to Semantic Textual Similarity (STS) tasks (Agirre et al., 2012), where we calculate the Spearman's correlation between models' predicted similarity and the *analogy scores* ($\alpha$) derived from annotations (§ 3.1). For

the second evaluation, we reframe our dataset as multiple-choice questions and evaluate LLMs on this set (§ 3.2).

## 3.1 Correlation with the *analogy score* $\alpha$

Similar to the STS-style evaluation (Agirre et al., 2012), we assess whether models can predict analogy scores based on embeddings (for encoder models) or by generation (for LLMs). We use a model to predict the similarity $f(\cdot, \cdot)$ for two stories. For encoder models, $f(s1, s2) =$ `Cosine(Encoder(s1), Encoder(s2))`. For LLMs, we prompt them to predict the `EntSim` and `RelSim` for the two stories. Finally, Spearman's correlations between the predicted similarity and the respective scores are reported.

**Setups.** We consider both encoder models and LLMs as baselines. Details are in § A.2.

The encoder models we evaluate include RoBERTa (Liu et al., 2019), SimCSE (Gao et al., 2021), OpenAI-ada (`text-embedding-ada-002`), Discourse Marker Representation (DMR) (Ru et al., 2023), RelBERT (Ushio et al., 2021b), and GloVe embeddings (Pennington et al., 2014) on nouns, verbs, or all words[10]. In addition to the unsupervised encoder models, we also fine-tune two models on the training set: a regression model, RoBERTa-Reg, which has a multilayer perceptron on top of the RoBERTa model that predicts `EntSim` and `RelSim`, and a contrastive learning-based model, RoBERTa-CL, which uses a contrastive learning objective to optimize its representations.

For LLMs, we test FlanT5 (Chung et al., 2022), LLaMa (Touvron et al., 2023), ChatGPT (OpenAI, 2022), and GPT-3.5 (`text-davinci-003`). Each model input is composed of three parts: the instructions, which give explanations to the similarity scores; $N$ examples, and the query story pair. We evaluate models with two instructions (short and long, where short instructions only contain the labels, and long instructions additionally have label definitions), and $N$ is set to 0, 1, or 3.

**Results.** The overall evaluation results are presented in Table 2. Generally, the models perform relatively poorly on the analogy score $\alpha$, indicating that there is still room for improvement on STORYANALOGY.

---

[10]We use Stanza (Qi et al., 2020) to conduct part-of-speech tagging for words.

| | ProPara | | | ROCStories | | | Word-Analogy | | | ConceptNet | | | Mean | | |
|---|---|---|---|---|---|---|---|---|---|---|---|---|---|---|---|
| | E | R | $\alpha$ | E | R | $\alpha$ | E | R | $\alpha$ | E | R | $\alpha$ | E | R | $\alpha$ |
| Random | 0.0 | 0.0 | 0.0 | 0.0 | 0.0 | 0.0 | 0.0 | 0.0 | 0.0 | 0.0 | 0.0 | 0.0 | 0.0 | 0.0 | 0.0 |
| Human | 54.9 | 64.6 | 67.9 | 82.9 | 70.1 | 55.2 | 67.1 | 69.4 | 58.3 | 53.7 | 75.5 | 68.5 | 64.7 | 69.9 | 62.5 |
| *Encoder models* | | | | | | | | | | | | | | | |
| RoBERTa | 45.2 | 41.9 | 6.2 | 20.6 | 22.2 | 7.6 | 34.8 | 24.5 | 0.9 | 34.9 | 28.8 | 9.8 | 33.9 | 29.4 | 6.1 |
| SimCSE | 48.0 | 38.4 | 1.8 | 14.4 | 12.7 | 2.6 | 43.2 | 26.8 | -2.0 | 30.7 | 21.2 | 3.7 | 34.1 | 24.8 | 1.5 |
| OpenAI-ada | 52.8 | 43.9 | 3.4 | 22.3 | 21.7 | 4.5 | 41.3 | 24.0 | -3.8 | 32.3 | 17.8 | -1.2 | 37.2 | 26.9 | 0.7 |
| DMR | 34.8 | 42.0 | 12.6 | 20.1 | 35.0 | 20.1 | 17.3 | 18.7 | 7.3 | 21.9 | 19.1 | 5.5 | 23.5 | 28.7 | 11.4 |
| RelBERT | 37.9 | 38.8 | 7.5 | 15.6 | 20.6 | 9.1 | 28.6 | 15.5 | -3.6 | 26.6 | 24.7 | 8.2 | 27.2 | 24.9 | 5.3 |
| GloVe-Noun | 35.2 | 18.5 | -7.8 | 9.4 | 6.9 | 2.5 | 29.8 | 14.2 | -5.6 | 27.7 | 13.0 | -2.2 | 25.5 | 13.2 | -3.3 |
| GloVe-Verb | 27.3 | 44.8 | 21.3 | 22.7 | 34.2 | 17.3 | 9.6 | 7.0 | 1.3 | 13.0 | 1.0 | -7.2 | 18.1 | 21.7 | 8.2 |
| GloVe-All | 36.3 | 29.0 | -1.0 | 28.7 | 27.2 | 4.8 | 26.1 | 12.3 | -5.1 | 18.6 | 3.3 | -7.9 | 27.4 | 18.0 | -2.3 |
| *LLMs* | | | | | | | | | | | | | | | |
| FlanT5-xxl | 41.9 | 21.8 | 4.8 | 9.4 | -8.7 | -2.4 | 40.0 | 27.5 | 8.3 | 37.7 | 26.7 | 8.1 | 32.3 | 16.8 | 4.7 |
| LLaMa-65B | 16.8 | 3.8 | -1.3 | 0.4 | -10.2 | -7.7 | 31.6 | 25.3 | 9.5 | 13.8 | 22.1 | 4.2 | 15.6 | 10.3 | 1.2 |
| GPT-3.5 | 24.1 | 11.4 | -6.4 | -3.2 | -2.8 | -6.4 | 34.2 | 26.9 | 8.6 | 28.8 | 30.0 | 7.9 | 21.0 | 16.4 | 0.9 |
| ChatGPT | 26.9 | 11.9 | -2.3 | 1.7 | -4.4 | -5.0 | 46.6 | 31.4 | 3.4 | 32.1 | 36.4 | 13.1 | 26.8 | 18.8 | 2.3 |
| *Finetuned models* | | | | | | | | | | | | | | | |
| RoBERTa-Reg | 38.5 | 34.8 | 16.6 | 14.5 | 26.2 | 12.0 | 20.1 | 28.8 | 19.4 | 23.8 | 32.0 | 20.1 | 24.2 | 30.5 | 17.0 |
| RoBERTa-CL | 25.7 | 53.9 | 35.2 | 29.1 | 47.2 | 28.3 | 33.8 | 40.9 | 21.1 | 26.0 | 30.8 | 15.3 | 28.7 | 43.2 | 25.0 |

Table 2: STS-style evaluation on different domains of STORYANALOGY. The values represent the Spearman's correlation (%) between the model prediction and scores from dataset (E, R, and $\alpha$). Here, E, R, and $\alpha$ correspond to `EntSim`, `RelSim`, and the analogy score `RelSim/EntSim`, respectively. The LLM performance is evaluated under the "long instruction+3-shot" setting.

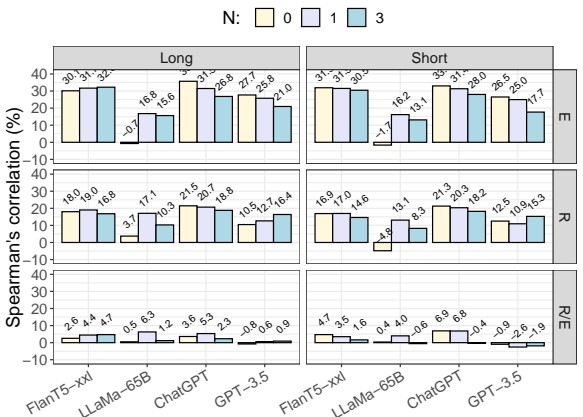

Figure 4: Spearman's $\rho$ (%) of LLMs, averaged across data domains. Here E, R, and R/E indicate `EntSim`, `RelSim`, and `RelSim/EntSim` ($\alpha$).

We have the following observations: (1) Similarities from state-of-the-art sentence embedding models are not good indicators for story analogy. Encoders such as RoBERTa, SimCSE, and OpenAI-ada show relatively good correlation with `EntSim` and `RelSim`, but they perform poorly on the analogy score $\alpha$. This suggests that their embeddings are suitable for literal similarity retrieval but not analogy retrieval. (2) Relational feature-aware models are better at analogy identification. Additionally, we find that encoder models aware of relation information, such as DMR (discourse relation), RelBERT (inter-word relation), and GloVe-

Verb (predicates), correlate better with the analogy score $\alpha$. (3) Finetuning improves models' analogy identification ability. The finetuned models, RoBERTa-Reg and RoBERTa-CL, are the top-performing models that significantly outperform all the other baselines on $\alpha$. (4) Generally, LLMs do not perform well on the analogy score $\alpha$. As shown in Figure 4, most LLMs can benefit from longer instructions as the extra definitions help in understanding the scores. Moreover, we find that despite its size, FlanT5-xxl is one of the best-performing LLMs in terms of predicting `EntSim` and `RelSim`.

## 3.2 Multiple choice evaluation

We construct a multiple-choice evaluation set using the annotated story pairs. First, we gather story pairs with `EntSim` < 1.0 and `RelSim` > 2.0. For each target story, we choose 3 negative choices to form the candidates. Out of these, two (easy) negative choices are randomly selected, while one (hard) negative example is chosen by retrieving stories with high nounal similarity (measured by the cosine similarity of the nounal GloVe embeddings) and < 50% token overlap. An example question is provided in Table 3. To assess human performance, we conduct human annotations.

We assess LLMs on multiple-choice questions. Each model input consists of an instruction, $N$ examples of multiple-choice questions, and the query

| Question: | Which candidate story is the best creative analogy for the source story? |
|---|---|
| Source: | Carbonic acid in rainwater breaks down rock. Plants grow in rock. |
| (0) | Plants and animals grow and reproduce. The population size gets larger and larger. |
| (1) | Recyclables are placed in a centralized container for the house. Recyclables are picked up by a recycling company. |
| (2) | Salty ocean water erodes metal. Corals thrive on metal. |
| (3) | The roots of the growing plants start to break up the rock. The plant acids dissolve the rock. |
| Answer: | (2) |

Table 3: An example of the multiple choice question. The goal is to select a candidate story that is the best analogy for the source story.

| Model | N-shot | | | Model | Question template | | |
|---|---|---|---|---|---|---|---|
| | 0 | 1 | 3 | | A | B | C |
| FlanT5-xxl | 45.0 | 46.3 | 45.0 | FlanT5-xxl | 41.2 | 47.1 | 48.0 |
| LLaMa-65B | 27.7 | 28.6 | 29.5 | LLaMa-65B | 29.6 | 26.0 | 30.2 |
| ChatGPT | 35.8 | 29.2 | 32.3 | ChatGPT | 30.1 | 33.3 | 33.9 |
| GPT-3.5 | 44.2 | 34.1 | 33.1 | GPT-3.5 | 33.7 | 33.6 | 44.0 |

Table 4: Multiple choice evaluation results. Each value represents the average accuracy (%) across different number of demonstrations (left) or across three question templates (right). The random and human performance are 25% and 85.7%, respectively.

multiple-choice question. We evaluate the models using three different instructions, such as "Which candidate story is the best creative analogy for the source story?", where **N** can be 0, 1, or 3. As a baseline, we obtain the performance of the analogy retrieval model in (Sultan and Shahaf, 2023) on our multiple choice questions, which achieves an accuracy of 44.9%.

**Results.** The results are presented in Table 4. Interestingly, while annotators can answer the questions correctly at an accuracy of 85.7%, LLMs struggle on selecting the most analogous story (the averaged accuracy for text-davinci-003 is merely 37.1%). Increasing the number of demonstrations does not show consistent benefits to model prediction. Also, we find that explicitly instructing models to choose the "creative analogy" (§ A.3, question template B) or to provide a definition of SMT when explaining analogies (template C) yields better performance compared to simply asking models to select the best analogy (template A).

We present the breakdown ratio of the percentage of types of choices selected in Table 5. We have the following observations: (1) LLMs can can differentiate between randomly sampled easy negatives and other choices. The proportion of easy neg-

| | Target | Hard | Easy |
|---|---|---|---|
| Random | 25.0 | 25.0 | 50.0 |
| (Sultan and Shahaf, 2023) | 44.9 | 17.8 | 37.2 |
| FlanT5-xxl | 45.4 | 37.2 | 17.4 |
| LLaMa-65B | 28.6 | 59.7 | 11.7 |
| ChatGPT | 32.4 | 59.5 | 8.1 |
| GPT-3.5 | 37.1 | 55.8 | 7.1 |

Table 5: Breakdown ratio (%) of model predictions: The table presents the percentage of different types of choices selected. "Target" refers to the ground-truth target, which is the analogous story. "Hard" and "Easy" refer to the negative examples sampled using nounal similarity and random sampling, respectively.

atives they select is less than 20%, whereas random chance would be 50%. Furthermore, more powerful LLMs like GPT-3.5 are better at this judgement compared to LLaMa and FlanT5-xxl. (2) LLMs can be easily distracted by hard negatives, as they often have a similar or higher chance of selecting hard negative choices instead of the targets. This suggests that the models prioritize surface similarity over structural similarity, despite the latter being more important in identifying analogies.[11] (3) In comparison, the baseline model from (Sultan and Shahaf, 2023) is more resilient against hard negative distractions. This is likely due to its framework design, which captures the structural similarity between stories by clustering entities and finding the mappings between clusters.

## 4 Story Analogy Generation

We examine whether the dataset STORYANALOGY can enhance the ability of analogy generation. We evaluate FlanT5 (Chung et al., 2022), LLaMa (Touvron et al., 2023), ChatGPT (OpenAI, 2022), and GPT-3.5 in zero-shot and few-shot settings using 40 source stories from the test set. To explore the potential of smaller models in generating high-quality analogies, we fine-tuned FlanT5-xl (3B parameters) and FlanT5-xxl (11B parameters) using the same template.

A crowd annotation is conducted to evaluate the quality of the generated stories from the models mentioned above. Workers are provided with a source story and its corresponding generated target story. They are then asked to assess the following: (1) Whether the target story is an analogy for the

---

[11]This phenomenon was also observed in visual analogies (Bitton et al., 2023), where they found that that models can solve visual analogies well when the distractors are random, but struggle with difficult distractors.

| Setting | Model | Generation quality | | |
|---|---|---|---|---|
| | | **Analogy** | **Novelty** | **Plausibility** |
| Zero | FlanT5-xl | 52.5 | 48.3 | 92.5 |
| | FlanT5-xxl | 46.7 | 49.2 | 92.5 |
| | LLaMa-65B | 38.3 | 39.2 | **93.3** |
| | ChatGPT | 70.0 | 72.5 | 90.8 |
| | GPT-3.5 | 75.8 | 81.7 | 87.5 |
| Few | FlanT5-xl | 48.3 | 50.0 | 91.7 |
| | FlanT5-xxl | 40.0 | 43.3 | 85.0 |
| | LLaMa-65B | 66.7 | 66.7 | 92.5 |
| | ChatGPT | **78.3** | **83.3** | 86.7 |
| | GPT-3.5 | 77.5 | 79.2 | 88.3 |
| Tuned | FlanT5-xl | 65.8 | 79.2 | 88.3 |
| | FlanT5-xxl | 72.5 | 81.7 | 86.7 |

Table 6: The crowd-annotated generation quality (%) in terms of (1) Whether the target story is considered an analogy to the source; (2) Novelty of the target story; (3) Plausibility of the generations.

source (as opposed to being a literal similarity or something else); (2) whether the target story is novel compared to the source; and (3) whether the target is plausible (More details can be found in § A.4). The average scores from three annotators are reported in Table 6. Example generations are shown in Table 7.

Under the zero-shot setting, we observe that FlanT5 and LLaMa struggle to generate meaningful analogies. They often tend to repeat patterns from the source stories (e.g., only replacing one word). In contrast, ChatGPT and GPT-3.5 produce more flexible stories that are frequently considered as analogies and novel.

Stories in STORYANALOGY can help models generate better analogies. With a few demonstrations, we observe a significant improvement in the generation quality of LLaMa (+28.4% and +27.5%). Moderate improvement on ChatGPT and GPT-3.5 is also observed. Notably, finetuning smaller LMs enhanced their generation quality. The finetuned FlanT5-xxl model performs better than the zero-shot ChatGPT and is comparable to the few-shot ChatGPT and GPT-3.5, despite having fewer parameters. Furthermore, while models become more creative through finetuning and in-context learning, their generation plausibility decreases, indicating an increase in hallucination.

## 5 Related Work

**Word-level analogy.** One of the famous works on word-level computational analogy was (Mikolov et al., 2013), where they found that word analogies can be predicted by word vector offsets. For instance, $\overrightarrow{King} - \overrightarrow{Man} + \overrightarrow{Woman} \approx \overrightarrow{Queen}$. With

| Source | A projectile is affected by gravity. It falls and picks up speed. |
|---|---|
| *Zero-shot* | |
| ChatGPT | Just as a sled sliding down a steep hill gains momentum as it accelerates, so does a projectile as it falls under the force of gravity. |
| GPT-3.5 | An Olympic runner is running a middle distance race. |
| LLaMa-65B | A projectile is affected by gravity. It falls and picks up speed. |
| FlanT5-xl | A rocket is affected by gravity. It falls and picks up speed. |
| FlanT5-xxl | A car is affected by gravity. It picks up speed and goes faster. |
| *Few-shot* | |
| ChatGPT | A seed is planted in fertile soil. It takes root and grows, flourishing into a healthy plant. |
| GPT-3.5 | An alpha particle is buffeted by the atmosphere. It spirals and eventually finds the ground. |
| LLaMa-65B | A projectile is affected by gravity. It falls and picks up speed. |
| FlanT5-xl | A rocket is affected by gravity. It falls and picks up speed. |
| FlanT5-xxl | A car is affected by gravity. It accelerates and picks up speed. |
| *Tuned* | |
| FlanT5-xl | A meteor is propelled by a tug. It moves and finds its way. |
| FlanT5-xxl | An idea is planted by brainstorm. It takes shape and comes to fruition. |

Table 7: Examples showing the source story and model generations under zero-shot, few-shot, and finetuning settings.

the development of pretrained language models (PLMs) such as BERT (Devlin et al., 2018), there have been works utilizing PLMs to solve word analogies by LM perplexity (Ushio et al., 2021a), pretrain relational embedding on certain prompt templates (Ushio et al., 2021b), or use word analogies as latent restriction to implicitly probe relational knowledge (Rezaee and Camacho-Collados, 2022).

In this line of work, a typical evaluation setting is ranking word pairs based on their relational similarity with the source pair (Mikolov et al., 2013; Czinczoll et al., 2022). For instance, given a word pair A:B, the aim is to select a target pair C:D such that the relation between C and D is the most similar to A:B among all candidates. This is similar to our multiple-choice evaluation setting.

In comparison, only a handful of research has been done in sentence or paragraph-level analogy:
**Analogous text retrieval.** Built on the famous structure mapping theory, SME (Falkenhainer et al., 1989) and LRME (Turney, 2008) model the analogy retrieval problem as an entity-mapping problem. They then solve this problem through web mining. Sultan and Shahaf (2023) develops a QA-SRL based analogy retrieval method to conduct

entity mapping. However, these works evaluate their methods by annotating the precision of the top-ranked results, leaving no large-scale analogy evaluation benchmarks to date.

**Analogy generation.** Recently, there have been attempts at pretraining or prompting LMs for analogy generation. Bhavya et al. (2022); Webb et al. (2022) evaluated LLMs' ability on solving word analogy tasks, where they found that large language models such as GPT-3.5 can surpass human performance on certain word analogy tasks. Ding et al. (2023) evaluated LLMs' creativity in terms of cross-domain analogies. Bhavya et al. (2022); Chen et al. (2022a) evaluated LMs' ability on generating explanations for word analogies. Bhavya et al. (2023) proposed a novel analogy mining framework based on generation.

**Analogy benchmarks.** There are many word-level analogy datasets. Google (Mikolov et al., 2013), BATS (Gladkova et al., 2016) contain relatively easier syntactic or shallow semantic relations. In contrast, U2 and U4, and Czinczoll et al. (2022) include examples with relatively more abstract relations. To the best of our knowledge, there is no large-scale story-level analogy data or resources as of the time of writing. The only related works here are (Li and Zhao, 2021; Zhu and de Melo, 2020), which transform word analogy pairs into sentence pairs with a few templates. Nagarajah et al. (2022) tried to annotate a tiny scale story analogy benchmark based on fables, but they failed to achieve. Wijesiriwardene et al. (2023) re-organized sentence relation datasets, where they viewed such relations (e.g., entailment, negation) are analogy, which is fundamentally different from our settings.

**Analogy in other domains.** In addition to analogies on word pairs and stories, there have been related studies on other topics. Hope et al. (2017) contribute a method for analogy mining over products. Chan et al. (2018) mine analogies from research papers with respect to their background, purpose, mechanism, and findings. Gilon et al. (2018) develop a search engine for expressing and abstracting specific design needs. Recently, Bitton et al. (2023) propose a visual analogies dataset, VASR, where they found that models struggle to find out analogies when given carefully chosen distractors.

## 6   Conclusion

We introduce STORYANALOGY, a multi-domain story-level analogy corpus with 24K story analo-

gies pairs annotated on two similarities under the extended SMT. To assess the analogy identification and generation capabilities of various models, we have devised a series of tests based on STORYANALOGY. The experimental findings indicate that current encoder models and LLMs still fall short of human performance in analogy identification. Additionally, we demonstrate that generative models can greatly benefit from our dataset.

## Limitations

We attempted to ensure dataset coverage by utilizing seed data from various sources. However, there are still specific domains that we were unable to include, such as biomedical stories or academic articles. We can extend the annotation to these domains using the annotation framework and evaluation metrics mentioned in this paper. Additionally, we have explored applications such as analogy identification (Section 3) and generation (Section 4). The potential of STORYANALOGY to be applied for creativity generation tasks (such as poetry, lyrics and humor generation) has not been fully investigated. Further development on other sources and applications is left as future work.

## Ethics Statement

The generated knowledge in STORYANALOGY have been carefully evaluated by crowd annotators to remove any possible toxic or counterfactual content. We set the threshold to be as low as 10%, such that any annotator's labeling an instance as toxic will lead to its removal. 142 instances are removed during this process. We conformed to recognized privacy practices and rigorously followed the data usage policy. We declare that all authors of this paper acknowledge the *ACM Code of Ethics* and honor the code of conduct.

## Acknowledgements

The authors of this paper were supported by the NSFC Fund (U20B2053) from the NSFC of China, the RIF (R6020-19 and R6021-20) and the GRF (16211520 and 16205322) from RGC of Hong Kong. We also thank the support from the UGC Research Matching Grants (RMGS20EG01-D, RMGS20CR11, RMGS20CR12, RMGS20EG19, RMGS20EG21, RMGS23CR05, RMGS23EG08).

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

# A  Appendix

## A.1  Details in creating STORYANALOGY.

### A.1.1  Details in generating candidates.

We prompt `text-davinci-003` model to collect the story analogy candidates. The prompt template is

**Demonstrations.** The in-context learning seed examples are presented in Table 8. In addition to the golden story analogies, we also curated the corresponding keyword pairs with regard to each story pair. This keyword pairs are useful for prompting to generate candidate stories from the Word Analogy and ConceptNet inputs, where their input data format are word-pairs (e.g. word: language :: note: music).

To construct a list of demonstrations for each data source, we ask experts to construct a set of analogous story pairs by web-searching and revising the results. Then, to ensure the diversity of the analogy, we make a list of orthogonal topics in each dataset, and randomly sampled demonstrations from these subtopics every time we construct a prompt.

**Prompt templates for "generating from story pairs".** The template for the demonstration is: "Example:\n(1){source story(i)}\nAn analogy for story (1) can be:\n(2){target story(i)}"

It is concatenated with a prompt at the end: "Example:\n(1){source story}\nAn analogy for story (1) can be:"

**Prompt templates for "generating from word pairs".** The prompt template for generating from word pairs is: "Write a group of 2-sentence

| Source story | Target story |
|---|---|
| The stream becomes a river. The river continues to flow along the same path for a long time. 
 ENTITY: stream, river | A person grows from a child into an adult. As time passes, the person experiences ongoing growth and maturation. 
 ENTITY: child, adult |
| Magma rises from deep in the earth. The magma goes into volcanos. 
 ENTITY: magma, volcanos | Food goes up from the stomach. The food enters the esophagus. 
 ENTITY: food, esophagus |
| The plasma membrane encloses the animal cell. It controls the movement of materials into and out of the cell. 
 ENTITY: plasma membrane, cell | Security guards monitor the doors of the factory. They manage the entry and exit of personnel to and from the factory. 
 ENTITY: security guard, factory |
| The tadpole begins storing food in the tail. The tadpole develops hind legs and lives off food stored in the it's tail. 
 ENTITY: tadpole, food | A person saves money in a savings account. The person relies on the saved funds to meet future financial obligations and sustain their lifestyle. 
 ENTITY: human, money |
| The sediment near the bottom is compressed by the weight of newer sediment. The sediment becomes sedimentary rock as it is pushed together by the heavy weight. 
 ENTITY: sediment, sedimentary rock | A person's ideas and beliefs are shaped by their experiences and influences. The person's thoughts and opinions become more solidified and defined as they are influenced by outside forces. 
 ENTITY: belief, solidified belief |
| Morgan enjoyed long walks on the beach. She and her boyfriend decided to go for a long walk. 
 ENTITY: beach, walking | Lenny liked to climb trees. He embarked on a tree-climbing expedition in the woods. 
 ENTITY: woods, climbing trees |
| He got a call from his girlfriend, asking where he was. Frank suddenly realized he had a date that night. 
 ENTITY: call, date | She received a notification on her phone, reminding her of an upcoming meeting. Jane suddenly remembered there was an important presentation to give. 
 ENTITY: notification, presentation |
| She was petrified and prayed to get out of the test. On the last day of lessons, the bus broke down and she was spared. 
 ENTITY: test, fear | He was terrified of the upcoming job interview. Due to oversleeping on the day of the interview, he missed the appointment and thus avoided the stress. 
 ENTITY: job interview, stress |
| He is only two weeks into his job and he is nervous. Every time he responds to calls he gets very worried. 
 ENTITY: job, nervous | Having recently started a relationship, she is grappling with anxiety. She becomes highly anxious whenever they have a disagreement. 
 ENTITY: relationship, anxious |
| She made sure she was quiet and respected others' space. It was strange that on Wednesday, she came to the office hung over. 
 ENTITY: introverted, getting drunk | James took care to comply with the rules and demonstrate deference towards authority figures. Surprisingly, he was caught shoplifting on a Friday. 
 ENTITY: disciplined, shoplifting |

Table 8: Seed analogy examples for generation candidates STORYANALOGY. We sample 5 pairs from propara and 5 paris from rocstories.

```
stories in around 30 words given the
keyword(s).\Hint:  Story 2 should be
analogous to story 1.\n Example 0:\n
Keywords for story 0: {keywords}"
```

In addition, we notice that the story pairs generated this way tend to have low entity similarity (since their entities are pre-given). Therefore, we additionally prompt LLMs to write a set of similar keywords for the source first, and then write a corresponding story, which does not need to be similar to the source.

```
"Write a group of 2-sentence stories in
around 30 words given the keyword(s).
Keywords 1: {}
Story 1: {}
Give a set of keywords similar to keywords
1, and then write a corresponding story
(the stories do not have to be similar):"
```

### A.1.2 Annotation templates.

In this section, we showcase our templates used for annotation on the Amazon Mechanical Turk platform. The instructions used for evaluating entity and relation similarities are presented in Figure 6 and Figure 8, respectively. Followed by these instructions, questions are presented using templates shown in Figure 7 and Figure 9.

### A.2 Details in § 3.1.

#### A.2.1 Baselines

**DMR**   Note that DMR requires two sentences as input. We use the NLTK toolkit to tokenize the stories into two sentences. In cases where there are not enough two sentences, we try to split the story by the first comma. If there is no comma in the story, the DMV vector is computed between the story and an empty string " ". This only accounts for a small number of stories (10-100).

**GloVe**   We use the glove-840B-300D version[12]. We first use the *Stanza* part-of-speech (PoS) annotation tool to parse the PoS of words. Then, we return the summation of embeddings for all the words with the corresponding PoS. Specifically, we determine nouns if a word's upos is in {"PROPN", "NOUN"}. We detect verbs if its upos is "VERB" or its xpos starts with "VB".

**RoBERTa-Reg**   We apply an MLP on top of the RoBERTa model and output two digits, which correspond to the EntSim and RelSim, respectively.

---

[12]https://nlp.stanford.edu/data/glove.840B.300d.zip

**RoBERTa-CL**   Details for training the contrastive learning model. We adopt the SimCSE training script[13] for training our contrastive learning model. The positive pairs are filtered according to the EntSim <= 1.0 and RelSim >= 2.0.

#### A.2.2 Prompt templates for similarity prediction.

The templates used for prompting LLMs to generate similarity predictions are presented below.

**The "long instruction" template for EntSim prediction.**

```
 "Evaluate the entity similarity between
a pair of stories. Assign the pair a score
between 0 and 3 as follows:
0  :    Unrelated.     The  two  stories
are talking about different topics and
entities of different types.
1 : Somewhat related.  The two stories
talk about different entities, and some
of them have similar or related types.
2 : Somewhat equivalent. The two stories
have different entities, but they have
the same types.
3 : Almost equivalent.   The entities
in the two stories are overlapped or
synonymous.
Following the above instruction, evaluate
the entity / topic similarity for S1 and
S2 (only answer by a score from 0, 1, 2,
3):
{N-DEMONSTRATIONS HERE} Q:
S1 - {INPUT-S1}
S2 - {INPUT-S2}
Score :"
```

**The "long instruction" template for RelSim prediction.**

```
"Evaluate the relation similarity between
a pair of stories.  Assign the pair a
score between 0 and 3 as follows:
0 : Very poor alignment. Most if not all
relationships do not align.
1 : Alignment with significant mismatches.
Some of the relationships align, but
there are some significant mismatches.
2  :    Alignment  with  insignificant
mismatches.  Most of the relationships
align  except  for  some  insignificant
mismatches.
3 : Alignment.   The relationships can
```

---

[13]https://github.com/princeton-nlp/SimCSE

```
align very well between the two stories.
Following the above instruction, evaluate
the relational similarity for S1 and S2
(only answer by a score from 0, 1, 2, 3):
{N-DEMONSTRATIONS HERE} Q:
S1 - {INPUT-S1}
S2 - {INPUT-S2}
Score :"
```

Here, we insert N$\in \{0, 1, 3\}$ demonstrations at the "N-DEMONSTRATION HERE" and fill in the story pairs at "INPUT-S1" and "INPUT-S2". The "short instruction" templates are similar, with the only difference that the detailed definition of scores is removed. For instance, "`0 : Unrelated. The two stories are talking about different topics and entities of different types.`" is replaced with "`0 : Unrelated.`"

### A.3 Details in § 3.2.

To construct the multiple-choice evaluation set, we gather story analogy pairs with `EntSim` < 1.0 and `RelSim` > 2.0. Next, we sample negatives for each story analogy pairs to form multiple choice questions. Similar to the GloVe baseline in § A.2, we obtain the nounal embedding for each story, and retrieve stories with high cosine similarities while have <50% overlapped tokens as the hard negative choices. We manually inspect the overall quality of the multiple choice questions constructed in this manner. We excluded the questions generated from the ROCStories split due to their lower quality, likely because the unusual distribution of `EntSim` in this split made it difficult to use the same method for creating the dataset as in the other splits (Figure 3). The resulting multiple choice dataset consists of 360 questions.

**Baselines in (Sultan and Shahaf, 2023)** We apply both the FMQ and FMV models, as suggested in (Sultan and Shahaf, 2023), to our story analogy identification task. To be precise, we gather the intermediate story pair similarities generated by their models. Afterwards, we choose the option that exhibits the highest similarity to the source story. Notably, the lengths of the stories in our dataset are considerably shorter than datasets used in its paper. Therefore, when running the baseline on our dataset, we adjusted the threshold of the similarity filter to better suit our settings. We selected a threshold of 0.3 for FMQ and 0.2 for FMV. For the other implementation details, we follow the origi-

nal settings in their code repo at https://github.com/orensul/analogies_mining. As for the result, we discovered that FMQ and FMV exhibited comparable performance (44.9% versus 44.7%) on the multiple choice dataset. The result from FMQ are reported in the main paper.

#### A.3.1 Prompt templates for multiple-choice evaluation.

The prompt template used in multiple-choice evaluation is:
```
" {QUESTION}
Source story: {}
Candidate stories:
(0): {}
(1): {}
(2): {}
(3): {}
Answer: "
```
where "QUESTION" is replaced with one of the following questions:
A: "`Select the candidate that best matches the source story as an analogy.`"
B: "`Which candidate story is the best creative analogy for the source story?`"
C: "`A creative analogy should have fewer similar entities but similar relational structures to the source story. Which candidate story is the best creative analogy for the source story?`"

### A.4 Details in the evaluation of story analogy generation.

#### A.4.1 Generation setups.

The models are evaluated under zero-shot, few-shot, and instruction-tuning settings. For zero-shot and few-shot prompting, the templates are: "`Write an analogy for story 1.\n \nStory 1: {}\nStory 2:`" This template is also used in the finetuning setting. For finetuning, we employ DeepSpeed[14] to accelerate the training on a single 8*V100 (32GB) instance.

#### A.4.2 Annotation.

We conduct crowd annotation on AMT to evaluate the generation quality. The annotation instruction is presented in Figure 5. During the annotation, the meta information of the target generation is hidden from the annotators and the requesters. In addition, the target stories are shuffled such that

---

[14]https://www.microsoft.com/en-us/research/project/deepspeed/

annotators cannot find out which models are used to generate the stories based on the order.

## A.5 Miscellaneous

In this section, we present some discussions that took place during the reviewing process.

### A.5.1 Potential applications of this work.

*Analogy Mining for Art and Design.* There have been various studies focusing on building analogical search engines. Hope et al. (2017) contribute a method for analogy mining over products. Chan et al. (2018) mine analogies from research papers with respect to their background, purpose, mechanism, and findings. Gilon et al. (2018) develop a search engine for expressing and abstracting specific design needs. Recently, Bitton et al. (2023) propose a visual analogies dataset, VASR, where they found that models struggle to find out analogies when given carefully chosen distractors. In computer graphics, some graphics design algorithms take as input an image from the user, and transform it to some other types of visual designs that are similar to the given image, such as embroidery patterns (Zhenyuan et al., 2023) and vector line arts (Mo et al., 2021). This category of work establishes connections between images and application-specific graphics patterns. With images as a guidance, the complicated visual design processes are made easy and intuitive for nonprofessional users.

*Analogical Reasoning.* Large language models (LLMs) have demonstrated impressive abilities in few-shot and zero-shot learning (Kaplan et al., 2020; OpenAI, 2022, 2023). Recently, ChatGPT (OpenAI, 2022), GPT-4 (OpenAI, 2023), Alpaca (Taori et al., 2023) and their following works (Chiang et al., 2023; Jiang et al., 2023) have achieved remarkable performance on a wide range of benchmarks. It is believed that they have acquired certain kind of analogical reasoning ability that are not only task-specific (Webb et al., 2022; Ding et al., 2023) , but also omnipresent throughout the prompting process of LLMs, and there are a lot of prompt engineering work to leverage this characteristic for downstream tasks (Jiang et al., 2022; Chan et al., 2023b,a,c; Chan and Chan, 2023). Meanwhile, it is important to note that LLMs also exhibit potential issues related to hallucination, biases, and privacy (Ray, 2023; Li et al., 2023a,b; Wang et al., 2023). Mitigating such issues often requires building up knowledge bases (Cheng et al.,

2021; Cui et al., 2021b,a), where analogy could be a useful angle to improve automatic building performance (Chen et al., 2022b). The data and evaluation metrics in this work may serve as a benchmark in evaluating one of the analogical reasoning abilities.

### A.5.2 Why the predictions of individual scores are good, but the prediction of $\alpha$ is bad.

Original question: *How is it that models are so good at individually predicting* EntSim *and* RelSim *(in Section 3.1), but they are not that good at predicting the analogy score $\alpha$?*

Since the analogy score is computed from both EntSim and RelSim, the prediction of the analogy score relies on predicting the gap between EntSim and RelSim, which is harder than predicting each similarity alone. A case is presented below to illustrate this.

Suppose we have four story pairs, and their ground-truth scores are: EntSim = [2, 1, 3, 0], RelSim = [0, 1, 2, 3].

The corresponding predictions are: EntSim'= [0, 2, 3, 0], RelSim'= [1, 0, 2, 1].

Then, the analogy scores and the predicted analogy scores can be computed from the above values (using $\alpha = \frac{\text{RelSim}}{\text{EntSim}+1}$)): $\alpha$ = [0, 0.5, 0.5, 3], $\alpha$' = [1, 0, 0.5, 1].

Finally, we can compute the Spearman's correlation coefficients as:

Corr(RelSim, RelSim')=0.316
Corr(EntSim, EntSim')=0.632
Corr($\alpha$, $\alpha$') = 0

Here, though the predictions of the respective scores have medium correlation with the ground-truths, the prediction of $\alpha$ has zero correlation.

Source story: **The virus invades cells. As a result, the DNAs are damaged.**

and several target stories:

Target story 1: **As virus infiltrates the body, it attacks healthy cells and causes a disturbance in the immune system.**

Target story 2: **Parasites invade their hosts, feed on nutrients, and weaken the immune system, causing damage to the host's body.**

Target story 4: **A swarm of space bees invaded the atmosphere and caused the planets to spin backwards. As a result, the gravitational pull of the universe was disrupted and time travel became possible.**

Your task is to answer three YES/NO questions for each target story:
**1. [Analogy?] Is the target story an analogy for the source story?**
Here, an analogy should have similar structures rather than details. For instance, target story 2 and 3 are analogies to the source story, while story 1 is not.

**2. [Novel?] Is the target story novel (compared to the source)?**
This measures the novelty of the target story. Think about whether the target is engaging and interesting to you. For instance, many may found target story 3 novel, while 1 and 2 not.

**3. [Plausible?] Is the target story plausible?**
Check whether the target story is plausible and makes sense. For instance, target story 4 does not really make sense and should be labeled as plausible.

Figure 5: The annotation instruction for generation quality evaluation.

| 3: Almost Equivalent | |
|---|---|
| Definition: The entities in the two stories are overlapped or synonymous. | |
| Toxins harm the body. The body tries to react to the toxins. | Poison enters the body. The body tries to fight off the poison. |
| Bogart lived on a farm. He loved bacon. | Mary lived on a farm. She loved bacon. |
| **2: Somewhat equivalent** | |
| Definition: The two stories have different entities, but they have the same types. | |
| John walked over to talk to her. Five minutes later he returned to a grill full of burned burgers. | Mary went inside to grab a drink. When she came back, the vegetables on the grill were charred. |
| Bogart lived on a farm. He loved bacon. | Jane lived in the city. She loved pizza. |
| Water vapor condenses. Clouds form. | Ice cubes form when water freezes. The water molecules become more organized and compact. |
| **1: Somewhat related** | |
| Definition: The two stories talk about different entities, and some of them have similar or related types. | |
| My wife went to the store to buy moving boxes. She bought boxes that were too large. | I went to the grocery store to buy fruit. I bought apples that were too ripe. |
| Put the aluminum can into a recycle bin. The cans are transported to a facility. | Put the dirty dishes into the dishwasher. The dishes are transported to the sink for cleaning. |
| **0: Unrelated** | |
| Definition: The two stories are talking about different topics and entities of different types. | |
| This liquid is known as magma. The magma rises to the earth's surface in volcanoes where it cools and hardens. | Security guards monitor the doors of the factory. They control the movement of people into and out of the factory. |
| The volcanos erupt many times. The size of the rocky area grows. | A plant grows larger over time. The plant's roots spread and deepen. |

Figure 6: The instructions used for evaluating entity similarity in human annotations.

| Source story | Target story | Scores 👨 | $\alpha$ | Domain |
|---|---|---|---|---|
| The stream becomes a river. The river continues to flow along the same path for a long time. | A person grows from a child into an adult. As time passes, the person experiences ongoing growth and maturation. | 🌼 : 0.6 💧 : 2.8 | 1.6 | PP |
| Fertilize the soil. Mix seeds into the fertilized soil. | Apply lotion to the skin. Massage the lotion into the skin. | 🌼 : 0.0 💧 : 2.6 | 2.6 | PP |
| Fill the tray with cool water. Place the tray in the freezer. | Fill the bucket with warm water. Place the bucket in the refrigerator. | 🌼 : 3.0 💧 : 3.0 | 0.8 | PP |
| The resulting material disappears. The plant becomes one with the soil. | The water evaporates. The liquid turns into vapor and disperses into the air. | 🌼 : 2.0 💧 : 0.4 | 0.1 | PP |
| The gas condenses in the condenser and becomes a liquid again. Heat is radiated away from the condenser. | An emotion is expressed and released. A calming effect follows after the expression. | 🌼 : 0.3 💧 : 0.0 | 0.0 | PP |
| They left him the key to the entrance. When Tom went over he realized it was the wrong key. | They gave her the password to the website. When Jane logged in, she realized it was the wrong password. | 🌼 : 1.0 💧 : 2.7 | 1.3 | ROC |
| I was building a dresser. I had several tools to help. | I was baking a cake. I had several ingredients to help. | 🌼 : 1.0 💧 : 3.0 | 1.5 | ROC |
| It's broken. I have to buy a new one. | It's expired. I have to get a new one. | 🌼 : 3.0 💧 : 3.0 | 0.8L | ROC |
| His cellmate tried to bully the man. The man fought his cellmate. | His classmate tried to intimidate him. The man stood his ground and refused to be bullied. | 🌼 : 2.7 💧 : 1.5 | 0.4 | ROC |
| The fight lasted until 10 am. We finally just went to bed out of exhaustion. | The argument went on until midnight. We eventually just gave up and went home in defeat. | 🌼 : 1.3 💧 : 0.0 | 0.0 | ROC |
| Foundations are poured to support the walls and roofs of buildings. The structure of the building is only as strong as it's foundation. | Reasons are formulated to make theories. The conclusions of theories are only as dependable as their initial premises. | 🌼 : 0.6 💧 : 1.8 | 1.1 | WA |
| The ground for the building is solid and secure. This gives the building its foundation and stability. | The reasons for the theory provide a rational explanation. This informs the decision-making process that supports the theories accuracy. | 🌼 : 0.4 💧 : 2.8 | 2.0 | WA |
| His memory has broken into fragmented pieces. He can recall flashes and images of the past, but nothing concrete or clear. | His memories remain a confused mess. Nothing holds together and what he remembers don't make sense. | 🌼 : 2.7 💧 : 3.0 | 0.8 | WA |
| Heat energy is transferred from one point to another. Transfers between different substances cause temperature changes. | Solid materials undergo phase transitions when energy is added. Changes in pressure can also result in phase transitions. | 🌼 : 2.8 💧 : 1.0 | 0.3 | WA |
| She laughed and let go of all of her worries. Her carefree attitude was liberating. | He delved into the unknown without a second thought, ignorant of the knowledge to come. | 🌼 : 0.8 💧 : 0.0 | 0.0 | WA |
| The student opens the book and begins to read. The knowledge gained from the book is absorbed by the student. | The cat sees a mouse and begins to chase it. The cat honing its hunting skills through practice and repetition. | 🌼 : 0.8 💧 : 1.4 | 0.8 | CN |
| The trigger is pulled and the pistol shoots. The gun fires. | A meteorite impacts Saturn's surface. The planet is buffeted by these larger objects. | 🌼 : 0.4 💧 : 2.8 | 2.0 | CN |
| He knew his only way out was to commit suicide. He was determined to die, no matter what. | She figured an overdose was the only way out. Within minutes, she had taken her last breath and died. | 🌼 : 3.0 💧 : 2.6 | 0.7 | CN |
| She purchased a round-trip ticket for her travels. She left with the assurance that she would return. | She chose her destination for her vacation with excitement. She anticipated what her journey would bring. | 🌼 : 2.8 💧 : 1.4 | 0.4 | CN |
| The rain began to pour and gradually, the river started to overflow. It was the start of a devastating flood. | The scissors snipped away, trimming her locks until her hair was just right. She became the proud owner of a new, short hairstyle. | 🌼 : 0.0 💧 : 0.0 | 0.0 | CN |

Table 9: Examples in STORYANALOGY with annotations from each domain. We report the EntSim 🌼 and RelSim 💧 from crowd workers 👨. The **Domain** column indicates the source of the story pairs. "PP", "ROC", "WA", and "CN" are short for "ProPara", "ROCStories", "Word Analogy", and "ConceptNet", respectively.

**Pair 1**

**Bogart lived on a farm. He loved bacon.**

**Jane lived in the city. She loved pizza.**

How will you rate the topic similarity between these two stories?

○ Almost equivalent! The entities in the two stories are overlapped or synonymous.

○ Somewhat equivalent! The two stories have different entities, but they have the same types.

○ Somewhat related. The two stories talk about different entities, and some of them have similar or related types.

○ Unrelated. The two stories are talking about different topics and entities of different types.

☐ At least one of these stories are ungrammatical or counterfactual.

Figure 7: The template for presenting a question regarding the evaluation of entity similarity in human annotations.

## Relational Similarity Rating

To determine the relational similarity, you are required to score it from 0 to 3. The higher the score, the more close two stories are.
In short, the definitions and examples of the four scores are provided below, first column is denoted as Story A and the second column contains Story B:

| 3: Perfect Alignment | |
|---|---|
| Definition: The relationships can align very well between the two stories. | |
| The animal's heart rate and breathing rate slow. The animal loses weight more slowly than usual. | The car's engine and exhaust system slow down. The car uses less fuel than usual. |
| The stream becomes a river. The river continues to flow along the same path for a long time. | A plant grows from a seed into a mature plant. The plant continues to grow and thrive in its environment over time. |
| Many more dead plants sink in the same area. The dead plants join together forming peat. | Many more leaves fall to the ground in the same area. The leaves pile up forming a layer of mulch. |
| **2: Alignment with insignificant mismatches** | |
| Definition: Most of the relationships align except for some insignificant mismatches. ("insignificant" means that the differences do not invalidate the relational similarity of the story pair.) | |
| Magma rises from deep in the earth. The magma goes into volcano. | Oxygen goes from the lungs to the blood. Blood goes to the heart.
(Explanation: In story B, the oxygen goes from the lungs to the heart through blood, which is slightly different from the relationship of story A.) |
| Sarah was on a bus to her work. She had to pee very badly. | Tom was driving to a client meeting. He suddenly realized he had a pressing need to use the bathroom.
(Explanation: The logical connection (inter-event/state relationship) between events can be slightly different. In story B, Tom "suddenly realized" the urgent need to go to the bathroom, while this is missing in story A.) |
| **1: Alignment with significant mismatches** | |
| Definition: Some relationships align, but there are some significant mismatches. | |
| The sound wave returns to the bat. The bat hears the echoed sound. | A person speaks to another person. The other person hears the words and responds.
(Explanation: Many event/state relationships in A cannot align with B.) |
| The cans are transported to a facility. The cans are shredded by a machine. | A package is delivered to a warehouse. The package is opened and its contents are sorted by workers.
(Explanation: The relationships in the underlined part cannot align with story A.) |
| Jo wanted to impress his friends. He went to a gator wrestling show. | Tom loved dancing. He went to a salsa club.
(Explanation: The relationships in the underlined part do not align.) |
| **0: Very poor alignment** | |
| Definition: Most if not all relationships do not align. | |
| Morgan enjoyed long walks on the beach. She and her boyfriend decided to go for a long walk. | I decided to take my girlfriend to the beach. As we were walking she paused. |
| This liquid is known as magma. The magma rises to the earth's surface in volcanoes where it cools and hardens. | Security guards monitor the doors of the factory. They control the movement of people into and out of the factory. |

Figure 8: The instructions used for evaluating relation similarity in human annotations.

**Pair 1**

**The animal's heart rate and breathing rate slow. The animal loses weight more slowly than usual.**

**The car's engine and exhaust system slow down. The car uses less fuel than usual.**

How will you rate the relational similarity between these two stories?

○ Perfect Alignment! The relationships can align very well between these two stories.

○ Alignment with insignificant mismatches. Most of the relationships align except for some insignificant mismatches.

○ Alignment with significant mismatches. Some relationships align, but there are some significant mismatches.

○ Very poor alignment. Most or even all relationships do not align.

☐ At least one of these stories are ungrammatical or counterfactual.

Figure 9: The template for presenting a question regarding the evaluation of relation similarity in human annotations.