# OpenReview forum: "StoryAnalogy: Deriving Story-level Analogies from Large Language Models to Unlock Analogical Understanding"
_EMNLP/2023/Conference — EMNLP 2023 Main_

### Official Review · Reviewer_2Qdd · 2023-08-01

**Soundness:** 4

**Excitement:**

4: Strong: This paper deepens the understanding of some phenomenon or lowers the barriers to an existing research direction.

**Paper Topic And Main Contributions:**

This paper presents a new dataset (called StoryAnalogy) which has around 24k annotations about analogies in context of stories.
The authors argue that the major work till now has been on word analogies, and work on sentences, and paragraph analogies has been limited.

With this background, the authors present StoryAnalogy in which authors do the following:
1. source stories from four diverse sources
2. use LLM to generate analogous stories corresponding to them.
3. get those stories annotated by humans

Further, authors show that current LLMs are very far behind humans in determining if two stories are analogies. They perform experiments on various tasks (for e.g: a) predicting similarity scores, b) selecting analogous stories amongst multiple options c) generating analogous stories ). The authors find that even the current best models are far behind humans on these tasks.


**Questions For The Authors:**

1. In section 3.1 for the encoder based baseline, lines 333-343 state that, analogous score was taken as cosine(encoder(s1), encoder(s2)). I am not convinced if similarity based on embeddings from just a pre-trained, off the shelf model would suffice. For example, I don't think that the scores for stories describing electrons and atoms & planets and sun would be close enough. Fine-tuning is necessary. I acknowledge that the authors present two fine-tuned models as well. I just think all other encoder models form a week baseline.
2. Section 3.1  lines 333-343 just describe how analogous score alpha would be calculated. They don't describe how entity similarity and relation similarity scores are calculated. Or did I miss it?
3. How is it that models are so good an individually predicting the two scores, but they are not that good at predicting the analogous score? And the fact that analogous score is such a simple combination of the other two makes is perplexing.
4. Appendix A.1 line 760. Is there something missing at the end of the line. " using the methods in '…' "
5. I think the the template mentioned in line 340 are not provided. Am I understanding correctly?


**Reasons To Accept:**

1. The paper introduces a great resource of analogous stories. They have around 24k story pairs annotated with scores for them being analogous.
2. The paper is very well written. The authors provide information about the prompts, dataset creation, human annotators in great detail. The main paper along with appendix seems well written and complete.
4. The authors follow a well thought of process of selecting annotators (Section 2.3)


**Reasons To Reject:**

1. It's not fully convincing that pre-trained off the shelf encoder models make strong enough baseline (Table 2). See the questions to author below for more details.
2. Since it's basically a dataset paper, it would have been nice to see datacards. (https://sites.research.google/datacardsplaybook/, https://arxiv.org/pdf/2204.01075.pdf )

**Reproducibility:**

4: Could mostly reproduce the results, but there may be some variation because of sample variance or minor variations in their interpretation of the protocol or method.

**Reviewer Confidence:**

4: Quite sure. I tried to check the important points carefully. It's unlikely, though conceivable, that I missed something that should affect my ratings.

---

> ### Author Rebuttal · Authors · 2023-08-29
>
> Thank you for reviewing our paper. We appreciate your thoughtful feedbacks and questions, and we hope the following discussions can help address your concerns:
>
> 1. [Reasons to reject 1, Question 1] It's not fully convincing that pre-trained off the shelf encoder models make strong enough baseline (Table 2).
>
> Using cosine similarity based on embeddings from off-the-shelf encoders is commonly used in semantic textual similarity, word analogy, and unsupervised discourse analysis tasks. To this end, we select representative encoder models from all kinds, such as word embedding model (GloVe), contextualized text embedding models (RoBERTa, SimCSE, OpenAI-ada), and contextualized text embedding models pretrained with relation-aware signals (RelBERT, DMR).
>
> Before the experiments, we cannot assume that the encoder models are weak baselines on our task. Also, these baselines help illustrate the relative performance improvement for our fine-tuned models. Therefore, we think that presenting results from these models would be necessary to draw conclusions.
>
> 2. [Reasons to reject 2] Since it's basically a dataset paper, it would have been nice to see datacards.
>
> Thank you for the suggestion. We found this a useful tool and we will try to add a datacard in the final version. At the current stage, we cannot present a datacard as we found it would reveal authors and affiliations and violate the anonymity requirement.
>
> 3.  [Question 2] Section 3.1 lines 333-343 just describe how analogous score alpha would be calculated. They don't describe how entity similarity and relation similarity scores are calculated. Or did I miss it?
>
> We are sorry for the misunderstanding caused. We will revise the corresponding part to make these points clear:
>
> (Line 339) For LLMs, we prompt them to predict the entity similarity and relation similarity scores for a given pair of stories, respectively. The prompt templates are presented in the response to Q5.
>
> (Line 338) For encoder models (including the finetuned RoBERTa-CL), we obtain their similarity prediction with the formula f(s1, s2) = Cosine(Encoder(s1), Encoder(s2)). We compute the correlation between f(s1, s2) and EntSim or RelSim or alpha, respectively.
>
> The finetuned regression model RoBERTa-Reg is coupled with an MLP layer that outputs both EntSim and RelSim.
>
> 4. [Question 3-1] How is it that models are so good an individually predicting the two scores, but they are not that good at predicting the analogous score?
>
> Since the analogy score is computed from both EntSim and RelSim, the prediction of the analogy score relies on predicting the gap between EntSim and RelSim, which is harder than predicting each similarity alone. Given that the prediction of EntSim and RelSim is far from perfect for the current baselines, the prediction of the analogy score is also unstable.
>
> We present a case here to illustrate this:
>
> EntSim = [2,1,3,0],
>
> RelSim = [0,1,2,3]
>
> EntSim-prediction = [0,2,3,0]
>
> RelSim-prediction = [1,0,2,1]
>
> The analogy scores can be computed from the above values (alpha=R/(E+1)):
>
> Alpha = [0.,  0.5, 0.5, 3. ]
>
> Alpha-prediction = [1.,  0.,  0.5, 1. ]
>
>
> The spearman’s correlation coefficients are:
>
> Corr(RelSim, RelSim-prediction) = 0.316
>
> Corr(EntSim, EntSim-prediction) = 0.632
>
> Corr(Alpha, Alpha-prediction) = 0.0
>
> Here, though the predictions of the respective scores have medium correlation with the ground-truths, the prediction of alpha has zero correlation.
>
> We will present a case study in the final version to help better understand this phenomenon.
>
>
> 5. [Question 4] Appendix A.1 line 760. Is there something missing at the end of the line. " using the methods in '…' "
>
> Yes. We will fix it in the final version. The related details are also mentioned at line 218. To facilitate constructing a set of demonstrations, we encouraged experts to retrieve information either by web search or browse through the provided story corpora.
>
> 6. [Question 5] I think the the template mentioned in line 340 are not provided. Am I understanding correctly?
>
> Thank you for pointing this out. We will add the templates to Appendix and release all the source codes in the final version. The templates are as follows:
>
>
>
> The “long instruction” template for entity similarity prediction (line 363):
>
> > Evaluate the entity similarity between a pair of stories. Assign the pair a score between 0 and 3 as follows:
> >
> > 0 : Unrelated. The two stories are talking about different topics and entities of different types.
> >
> > 1 : Somewhat related. The two stories talk about different entities, and some of them have similar or related types.
> >
> > 2 : Somewhat equivalent. The two stories have different entities, but they have the same types.
> >
> > 3 : Almost equivalent. The entities in the two stories are overlapped or synonymous.
> >
> > Following the above instruction, evaluate the entity / topic similarity for S1 and S2 (only answer by a score from 0, 1, 2, 3):
> >
> > {N-DEMONSTRATIONS HERE}
> >
> > Q :
> >
> > S1 - {INPUT-S1}
> >
> > S2 - {INPUT-S2}
> >
> > Score :
>
>
>
> The “long instruction” template for relation similarity prediction:
>
> > Evaluate the relation similarity between a pair of stories. Assign the pair a score between 0 and 3 as follows:
> >
> > 0 : Very poor alignment. Most if not all relationships do not align.
> >
> > 1 : Alignment with significant mismatches. Some of the relationships align, but there are some significant mismatches.
> >
> > 2 : Alignment with insignificant mismatches. Most of the relationships align except for some insignificant mismatches.
> >
> > 3 : Alignment. The relationships can align very well between the two stories.
> >
> > Following the above instruction, evaluate the relational similarity for S1 and S2 (only answer by a score from 0, 1, 2, 3):
> >
> > {N-DEMONSTRATIONS HERE}
> >
> > Q :
> >
> > S1 - {INPUT-S1}
> >
> > S2 - {INPUT-S2}
> >
> > Score :
>
>
>
> We insert N (N=0, 1, 3) demonstrations at the “N-DEMONSTRATION HERE” and input the story pairs. The“short instruction” templates has similar format to the above, except that the detailed definition of scores is removed, e.g. “0 : Unrelated. The two stories are talking about different topics and entities of different types.” is replaced with “0 : Unrelated.”

---

### Official Review · Reviewer_WoNd · 2023-08-08

**Soundness:** 4

**Excitement:**

4: Strong: This paper deepens the understanding of some phenomenon or lowers the barriers to an existing research direction.

**Missing References:**


Joel Chan, Joseph Chee Chang, Tom Hope, Dafna Shahaf, and Aniket Kittur, SOLVENT: A Mixed Initiative System for Finding Analogies between Research Papers
CSCW 2018

Karni Gilon, Felicia Ng, Joel Chan, Dafna Shahaf and Aniket Kittur, Analogy Mining for Specific Design Needs
CHI 2018

Tom Hope, Joel Chan, Aniket Kittur and Dafna Shahaf, Accelerating Innovation Through Analogy Mining
KDD 2017

VASR: Visual Analogies of Situation Recognition
Yonatan Bitton, Ron Yosef, Eliyahu Strugo, Dafna Shahaf, Roy Schwartz, Gabriel Stanovsky
AAAI 2023

**Paper Topic And Main Contributions:**

This paper introduces STORYANALOGY, a new dataset and set of benchmarks for evaluating story-level analogical reasoning in artificial intelligence systems. The key points are:

- Story-level analogies involve comparing entire narratives or sequences of events, which is more complex than just comparing individual words or concepts as in traditional analogy datasets. However, there has been little research on modeling story analogies computationally.

- The paper creates a large dataset of over 24,000 story pairs labeled for analogy using crowdsourcing. The analogies cover diverse domains like science, social narratives, and commonsense knowledge.

- To systematically assess different levels of analogy, the authors extend the cognitive psychology theory of Structure Mapping to define two similarity scores - entity similarity and relational similarity. These scores characterize how much the stories align in terms of entities/topics and relational structure.

- Using the dataset, the paper evaluates several state-of-the-art models (encoders like SimCSE and LLMs like ChatGPT) on story analogy identification and generation. The models significantly underperform humans, showing there is substantial room for progress.

- The dataset and benchmarks enable new research on improving AI's analogical reasoning over narratives. This could enable better commonsense understanding and transfer learning.

In summary, the key contribution is creating resources to advance research on an important but relatively under-studied AI capability - story-level analogical reasoning. The empirical results show current models still have significant limitations on this task.

**Questions For The Authors:**

N/A

**Reasons To Accept:**

Overall, this paper makes a solid contribution in terms of a new dataset, evaluation methodology, and empirical analysis on an important NLP problem. It should lead to increased community interest in complex analogical reasoning and encourage follow-up work leveraging the introduced resources. The novel dataset and experiments are also presented rigorously. This can positively impact the field by spurring research that could lead to AI systems with stronger narrative understanding, commonsense reasoning, and transfer learning abilities.

1) Addresses an important but under-studied problem in AI - modeling complex story-level analogies requires deeper semantic understanding and abstraction than traditional word or sentence-level analogy tasks.
2) Provides novel and sizable resources to the community - the dataset of over 24K annotated story analogy pairs enables new research in this direction.
3) Presents comprehensive empirical results and analysis - the authors thoroughly evaluate strong baseline models from encoder methods to large language models, clearly showing current limitations.
4) Promotes an evaluation methodology - the proposed analogy identification and generation tests based on structure mapping theory provides a principled way to assess progress.
5) Demonstrates clear value of the resource - the experiments show models can benefit from the dataset through fine-tuning and few-shot learning.
6) Well-written and structured - the background, dataset creation, experiments, and analysis are presented clearly.


**Reasons To Reject:**

I dont see any reasons to reject

**Reproducibility:**

5: Could easily reproduce the results.

**Reviewer Confidence:**

4: Quite sure. I tried to check the important points carefully. It's unlikely, though conceivable, that I missed something that should affect my ratings.

---

> ### Author Rebuttal · Authors · 2023-08-29
>
> Thank you for reviewing our paper! We appreciate your feedback on this work, and we will add the missing references in the revision.

---

### Official Review · Reviewer_yNU4 · 2023-08-08

**Soundness:** 3

**Excitement:**

3: Ambivalent: It has merits (e.g., it reports state-of-the-art results, the idea is nice), but there are key weaknesses (e.g., it describes incremental work), and it can significantly benefit from another round of revision. However, I won't object to accepting it if my co-reviewers champion it.

**Paper Topic And Main Contributions:**

The paper presents a method for generating analogical sentence pairs using extensive, unlabeled narrative corpora and large language models. Starting with inputs sampled from scientific narratives, stories, and knowledge graph tuples (e.g., ConceptNet), the authors employ ChatGPT (text-davinci-003) to produce analogies, aided by a set of seed paired data. These generated analogies are evaluated through crowd annotation, utilizing novel metrics like EntSim, RelSim, and alpha, inspired by the Structure Mapping Theory. The resulting dataset of story analogies is established and validated, while the authors also propose diverse techniques to fine-tune and test several Large Language Models (LLMs) and language encoders for the identification and generation of analogies.

Strengths:
- Interesting approach and a valuable dataset creation for exploring and applying analogical reasoning (a powerful AI paradigm) to NLP models.
- Clear annotation guidelines to evaluate automatically generated datasets
- Extensive evaluation of LLMs and encoders on the curated dataset, demonstrating their capabilities and limitations

Weaknesses:
- Essential details are not presented in and moved to the supplementary section: The crux of the work lies in synthetically generating sentence-analogy pairs using LLMs, but I had to go to Appendix to know the exact implementation details. Even the name of the LLM was not mentioned in the main section.
- Using a proprietary tool for analogy generation: Since, it’s not mentioned, I assumed that authors used text-davinci-003 in ChatGPT api to generate analogies. This is a proprietary system and not a lot has been disclosed about how it operates and what kind of pre- and post-processing steps it applies to get better generations. Additionally, proprietary systems like this change too fast, making it harder to replicate this work.
- I did not fully understand why “alpha” which is a ratio of RelSim and EntSim is a better metric to evaluate analogies. While it may mark high RelSim and low EntSim cases better analogies, it rules out other possibilities (such as entities paradigmatically related to each other, thus having a higher EntSim but perfectly making a good analogy).
- The font size in figures and tables are sometimes too small and illegible. Moreover, Figures and tables are not explained enough in sections. For example, what is the summary of figure 3 and what does it entail? Any interesting insights?
- Neither the paper not the ethics statement includes any potential shortcoming the dataset may have in terms of generating analogies that have some form of societal bias (pertaining to race, gender and occupation). At-least a detailed ethics statement should be provided.

In general, the dataset holds some value, yet there's room for presentation enhancement. The paper lacks clarity and justification in some sections while excessively elaborating on others. Resolving these matters would require significant revision, hence, my inclination is toward rejecting the current version.


**Questions For The Authors:**

Please address the weaknesses in the author response.

**Reasons To Accept:**

Interesting dataset, thorough qualitative and quantitative evaluation

**Reasons To Reject:**

The presentation and flow could e improved a lot. See the weaknesses above.

**Reproducibility:**

2: Would be hard pressed to reproduce the results. The contribution depends on data that are simply not available outside the author's institution or consortium; not enough details are provided.

**Reviewer Confidence:**

4: Quite sure. I tried to check the important points carefully. It's unlikely, though conceivable, that I missed something that should affect my ratings.

---

> ### Author Rebuttal · Authors · 2023-08-29
>
> Thank you for your feedbacks on this work! We hope the following discussions are helpful for addressing your concerns:
>
> 1. Essential details are not presented in and moved to the supplementary section.
>
> Due to the limited space, we put details (such as the prompt templates and the model we use) into the appendix section. The name of the LLM is mentioned at line 747 in Appendix A.1. (text-davinci-003). For the final version with one more page, we will definitely move important details to the main section.
>
> 2. Using a proprietary tool for analogy generation.
>
> Thank you for bringing up this point. In order to minimize potential misunderstandings, we are adding further specifications as follows:
>
> We would like to point out that which exact model to use to generate the data does not make a huge difference here (the LLM we use is text-davinci-003, at line 747), since we are presenting the annotated data, and that the data has been annotated and filtered by human workers. The generation model will not be used again after the candidate generation step.
>
> For replication purposes, our data will be made public in the final version. Additionally, the expansion of our data is possible because the candidate creation process and annotation guidelines have been clearly defined.
>
> 3. I did not fully understand why “alpha” which is a ratio of RelSim and EntSim is a better metric to evaluate analogies.
>
> As illustrated in the paragraph at line 143, according to the original Structure-Mapping Theory, analogy between objects holds when they have similar relational structures while dissimilar attributes, while “literal similarity” holds when both the relational structures and attributes are similar. In this work, we extend this to story level, which means story analogies should have similar relational structures (high RelSim) and dissimilar in terms of entity or topic (low EntSim).
>
> The analogy score “alpha=RelSim/EntSim” is an instantiation of such a notion to reflect the level of analogy between a pair of stories. There are of course other ways of implementing this, such as “RelSim-EntSim”. In practice, we conducted some preliminary experiments and found these definitions do not make a big difference, so we finally decided to use the current definition.
>
> We are not sure what it means by “entities paradigmatically related to each other”. (Could you please elaborate? Thanks!) If it means entities with close meanings, such as “Magma-Lava”. Then, according to the structure-mapping theory, such high EntSim would make it likely to be a literal similarity case (see Figure 2), and this is less desirable than analogy (“Magma-food”) which have lower EntSim.
>
> 4. The font size in figures and tables are sometimes too small and illegible. Moreover, Figures and tables are not explained enough in sections. For example, what is the summary of figure 3 and what does it entail? Any interesting insights?
>
> Thank you for pointing this out. We will adjust the font size of figures and tables to make them more friendly to readers.
>
> Figure 3 presents the distribution of annotations of EntSim and RelSim across different sources. From the plot, we can find that candidate pairs generated from the ROCStories have relatively high EntSim and RelSim compared to the other sources, which might be because ROCStories are a social commonsense story corpora that are mostly talking about similar social stories. This implies that genres of story source have influence on the score distribution of analogy generated.
>
> We will further add explanations to figures and tables.
>
> 5. Neither the paper not the ethics statement includes any potential shortcoming the dataset may have in terms of generating analogies that have some form of societal bias (pertaining to race, gender and occupation). At-least a detailed ethics statement should be provided.
>
> During the annotation (section 2.3, line 261, 294; also, in ethics statement at line 572), we required workers to label a story pair as “poor quality” if they find any toxic content. Story pairs with >10% annotators annotated with this tag are filtered out. This is a rather general instruction to remove the potentially toxic contents, so it is assumed that most possibly toxic contents have been removed in the final dataset (which accounts for 142 instances as mentioned at line 296). We will include a more detailed discussion and ethics statements on these toxic contents in the final version.
>
> Thank you again for the valuable suggestions. We will address these concerns and incorporate related discussion on them in the revision to improve the presentation and flow.

---

### Official Review · Reviewer_8L5d · 2023-08-10

**Soundness:** 4

**Excitement:**

3: Ambivalent: It has merits (e.g., it reports state-of-the-art results, the idea is nice), but there are key weaknesses (e.g., it describes incremental work), and it can significantly benefit from another round of revision. However, I won't object to accepting it if my co-reviewers champion it.

**Paper Topic And Main Contributions:**

Topic: The paper presents an in-depth exploration of story-level analogies in the context of natural language understanding by introducing an analogy corpus aimed at enhancing the performance of LLMs in multi-domain story-level analogy generation.

Main Contributions:

1.	Creation of the STORYANALOGY Corpus: The paper introduces a novel, large-scale corpus named STORYANALOGY, which comprises 24,000 pairs of story analogies from diverse domains, enriched with human annotations.

2.	Development of an Evaluation Framework: The authors devise tests and evaluations to assess the identification and generation of story-level analogies, utilizing an extended version of the Structure-Mapping Theory (SMT).

3.	Analysis of Performance: The paper uncovers the challenges faced by both sentence embedding models and large language models (LLMs) when performing story-level analogical tasks. Special emphasis is given to models such as ChatGPT and LLaMa.

4.	Introduction of Improvement Techniques: The paper demonstrates that the inclusion of STORYANALOGY data can enhance the quality of analogy generation in models, particularly through fine-tuning with specific models like FlanT5-xxl.


**Reasons To Accept:**

1.	Novelty: The approach to story-level analogies is unique and addresses a gap in existing research that has mostly focused on word-level analogies.

2.	Rich Data Resource: The creation of a substantial corpus specific to story analogies provides an invaluable resource for further research and development.

3.	Detailed Evaluation: The paper offers a comprehensive assessment of current models' capabilities, providing insights into where they succeed and where they fall short.

4.	Potential for Further Research: The insights and resources presented in this paper could stimulate further research in areas like creativity-based generation tasks, including poetry, lyrics, and humor generation.


**Reasons To Reject:**

1.	Restricted Domain Scope: Despite the utilization of diverse sources in assembling the corpus, certain specialized fields such as biomedical narratives or scholarly articles remain outside its coverage. This exclusion may constrict the corpus's range of applicability. Furthermore, the aggregate size of the corpus might be inadequate for the comprehensive fine-tuning of a large language model (LLM) with numerous parameters.

2.	Disparity in Model Performance: The paper accentuates a significant divergence in performance levels between existing models and human abilities. However, it fails to visually represent this gap succinctly in Section “3.1 Correlation with the analogy score α,” nor does it elucidate detailed strategies to mitigate this discrepancy.

3.	Constrained Evaluation Methodology: The automated assessments employed within this paper hinge on encoder models introduced from 2014 through recent years. With the exception of the newly publicized OpenAI-ada, the inclusion of models from previous years might not be methodologically sound for side-by-side comparison, as illustrated in Table 2.

4.	Absence of Defined Practical Applications: The paper could refine the exposition of its practical applications and the specific contexts where the methods and discoveries might be deployed. Given the breadth and complexity of tasks that an LLM can undertake, the provision of a multi-constrained, task-specific corpus may lack novelty or sufficient contribution to the enrichment of diversity or the enhancement of LLMs' overall performances.


**Reproducibility:**

3: Could reproduce the results with some difficulty. The settings of parameters are underspecified or subjectively determined; the training/evaluation data are not widely available.

**Reviewer Confidence:**

4: Quite sure. I tried to check the important points carefully. It's unlikely, though conceivable, that I missed something that should affect my ratings.

---

> ### Author Rebuttal · Authors · 2023-08-29
>
> Thank you for your thoughtful review of our paper. We appreciate your feedback on our work and would like to address your comments below:
>
> 1. Restricted Domain Scope.
>
> We agree that such extension will increase the value of our dataset, and we will discuss it in the Future Work section. In the future, using our annotation framework and evaluation metrics, we can extend the annotation to other domains, such as biomedical and scholarly articles, in addition to the four data domains in this study.
>
> 2. Disparity in Model Performance.
>
> Thank you for the suggestion on the visualization. We will emphasize the gap on the table and add visualization figures in the final version.
>
> In terms of strategies to mitigate the discrepancy, we have shown that:
>
> (1)   (At line 380) For unsupervised encoder models, we found that models aware of relation information in their pre-training stage, such as DMR, RelBERT, and GloVe-Verb have better correlation performance than the general ones.
>
> (2)   (At line 352 and line 384) By fine-tuning encoder models with regression and contrastive learning objectives on the training set, the model can improve substantially.
>
> (3)   (At line 388) For LLMs, we also conduct experiments on using different prompting setups (Figure 4), where we found that using extra definitions as hints in the prompt can help LLMs improve their prediction.
>
> These findings are useful for mitigating the discrepancy for different models. We will leave more detailed exploration for future work.
>
> 3. Constrained Evaluation Methodology.
>
> Since story level analogy has not been well-studied yet, the goal of the experiment shown in Table 2 is to evaluate whether the embeddings generated by existing encoder models can perform well on analogy identification. Therefore, the methodologies behind the models have little influence on the conclusions we made: We can treat the encoder model as a black box. Story texts are input to the black box, and we use its output embeddings for evaluation. We select representative encoder models from all kinds, such as word embedding model (GloVe), contextualized text embedding models (RoBERTa, SimCSE, OpenAI-ada), and contextualized text embedding models pretrained with relation-aware signals (RelBERT, DMR).
>
> Therefore, OpenAI-ada still makes a fair side-by-side comparison with the other models. In fact (Table 2), we found that although OpenAI-ada performs well on EntSim and RelSim correlations respectively, it has poor performance in terms of correlation with alpha, which is even worse than the DMR model (which is of the same size of RoBERTa-base). This indicates that it is not a good choice for analogy identification.
>
>   4. Absence of Defined Practical Applications.
>
> Thank you for the suggestion. We will make clear the practical applications of this work in the final version.  In our experiments, we have done applications such as analogy identification (in Sec 3) and generation (in Sec 4). In practice, analogy identification is related to applications such as creativity search engine, where the similarity scores (line 336) can be directly used for analogy passage retrieval. Analogy generation can be used for generating useful ideas or poems. There are many related works on these applications such as [Sultan and Shahaf, 2023], [Turney, 2008], [Ding et al., 2023].
> Moreover, we are working on a demo of these applications, which will come out in the final version of the paper. We hope these efforts can help illustrate the potential practical applications.
>
>  Thank you again for your constructive feedbacks. We will address these concerns and incorporate related discussion on them in the revision.

---

### Meta-Review · Area_Chair_ANkk · 2023-09-12

**Recommendation:** 4

**Metareview:**

The paper addresses an important and under-studied NLP problem. The main contributions of this work are a substantially novel large dataset of story analogies created by crowdsourcing, a valuable evaluation framework to assess the identification and generation of story-level analogies, a method for enhancing analogy generation which makes use of the novel dataset and fine-tuning with specific models like FlanT5.  The dataset is used experimentally to benchmark the performance of several LLMs; results highlight the task is still challenging both for sentence embedding models and LLMs.  Overall, the work has the potential to help advance research in story-level analogical reasoning. However, the presentation could be improved (see reviewer 2 comment about the imbalance in sections). Although some reviewers consider as limitations the lack of coverage of some domains, my view is that the current version of the work still sufficiently supports the authors’ claims. It is recommended that authors’ incorporate all suggestions by reviewers, as also accounted for in rebuttals. Especially, “any details important for understanding the key aspects of the work should be in the paper rather than in appendices, as per ACL reviewing policies.

**Pros.**

- The described approach to story-level analogies is novel and addresses a gap in existing NLP/AI research;

- The paper is generally well written, and the argumentation is solid, but there is some imbalance in details among sections;

- It states clearly its claims and goals and provides details about the prompts and the dataset creation process;

- The dataset is extremely valuable and can stimulate future research on the task of complex analogical reasoning;

- The annotation process is solid and carefully thought-through;

- The paper presents extensive empirical results in evaluation of current models' capabilities and shows how tested models can benefit from the newly produced dataset;

- The work has potential for future research.

**Cons:**

- Important implementation details are given only in Appendix;

- A proprietary tool is used for analogy generation. While this AC tends to agree with authors that this does not weaken the claims nor the results, it hampers full replicability;

- The dataset might be too small for an effective fine-tuning of LLMs;

- Missing justification/explanation of the use of α as the metric for evaluation and cornerstone for analysis;


**Other revisions needed:**

- Fix font issues in figures and tables;

- Better explain Figures and tables;

---

### Decision · Program_Chairs · 2023-10-07

**Decision:**

Accept-Main

**Comment:**

The paper addresses an important and under-studied NLP problem. The main contributions of this work are a substantially novel large dataset of story analogies created by crowdsourcing, a valuable evaluation framework to assess the identification and generation of story-level analogies, a method for enhancing analogy generation which makes use of the novel dataset and fine-tuning with specific models like FlanT5.  The dataset is used experimentally to benchmark the performance of several LLMs; results highlight the task is still challenging both for sentence embedding models and LLMs.  Overall, the work has the potential to help advance research in story-level analogical reasoning. However, the presentation could be improved (see reviewer 2 comment about the imbalance in sections). Although some reviewers consider as limitations the lack of coverage of some domains, my view is that the current version of the work still sufficiently supports the authors’ claims. It is recommended that authors’ incorporate all suggestions by reviewers, as also accounted for in rebuttals. Especially, “any details important for understanding the key aspects of the work should be in the paper rather than in appendices, as per ACL reviewing policies.

**Pros.**

- The described approach to story-level analogies is novel and addresses a gap in existing NLP/AI research;

- The paper is generally well written, and the argumentation is solid, but there is some imbalance in details among sections;

- It states clearly its claims and goals and provides details about the prompts and the dataset creation process;

- The dataset is extremely valuable and can stimulate future research on the task of complex analogical reasoning;

- The annotation process is solid and carefully thought-through;

- The paper presents extensive empirical results in evaluation of current models' capabilities and shows how tested models can benefit from the newly produced dataset;

- The work has potential for future research.

**Cons:**

- Important implementation details are given only in Appendix;

- A proprietary tool is used for analogy generation. While this AC tends to agree with authors that this does not weaken the claims nor the results, it hampers full replicability;

- The dataset might be too small for an effective fine-tuning of LLMs;

- Missing justification/explanation of the use of α as the metric for evaluation and cornerstone for analysis;


**Other revisions needed:**

- Fix font issues in figures and tables;

- Better explain Figures and tables;